# Hand Rehabilitation Devices: A Comprehensive Systematic Review

**DOI:** 10.3390/mi13071033

**Published:** 2022-06-29

**Authors:** Ryan Kabir, Md Samiul Haque Sunny, Helal Uddin Ahmed, Mohammad Habibur Rahman

**Affiliations:** 1Department of Mechanical Engineering, BioRobotics Lab, University of Wisconsin-Milwaukee, Milwaukee, WI 53211, USA; helal.uddin@tmtgbd.com (H.U.A.); rahmanmh@uwm.edu (M.H.R.); 2Department of Computer Science, BioRobotics Lab, University of Wisconsin-Milwaukee, Milwaukee, WI 53211, USA; msunny@uwm.edu

**Keywords:** hand rehabilitation, rehabilitation therapy, actuation mechanism, control system

## Abstract

A cerebrovascular accident, or a stroke, can cause significant neurological damage, inflicting the patient with loss of motor function in their hands. Standard rehabilitation therapy for the hand increases demands on clinics, creating an avenue for powered hand rehabilitation devices. Hand rehabilitation devices (HRDs) are devices designed to provide the hand with passive, active, and active-assisted rehabilitation therapy; however, HRDs do not have any standards in terms of development or design. Although the categorization of an injury’s severity can guide a patient into seeking proper assistance, rehabilitation devices do not have a set standard to provide a solution from the beginning to the end stages of recovery. In this paper, HRDs are defined and compared by their mechanical designs, actuation mechanisms, control systems, and therapeutic strategies. Furthermore, devices with conducted clinical trials are used to determine the future development of HRDs. After evaluating the abilities of 35 devices, it is inferred that standard characteristics for HRDs should include an exoskeleton design, the incorporation of challenge-based and coaching therapeutic strategies, and the implementation of surface electromyogram signals (sEMG) based control.

## 1. Introduction

Victims of cardiovascular diseases often experience deterioration of their motor function [1]. Two main impairments of motor function loss are spasticity and weakness, decreasing the patient’s quality of life and overall recovery [2,3]. Spasticity is defined as jerking motions caused by hyper-excitability in the stretch reflex [2,4]. Weakness is defined by the inability to complete extensions of the muscles. Muscular weakness is the primary contributor to motor function loss in the finger post-stroke [5]. Rehabilitation therapy helps to alleviate the obstacle and strengthen the muscles affected. Rehabilitation therapy is an integral process in recovery after losing motor function, especially for post-stroke victims [6]. Proper motor function recovery requires neural plasticity, the biological process of reforming neural pathways [7]. Neuroplasticity occurs in three stages [8]. First, cell death occurs to unmask new and secondary neuronal networks, thus returning the cortical pathways from inhibitory to excitatory [8,9]. Eventually, the neural networks regrow the connections with neuronal proliferation and synaptogenesis [10]. Through extensive studies, it has been proven that activity, exercise, and neural stimulation have promoted neural plasticity in patients [11,12]. Rehabilitation therapy is critical for patients’ improvement and assistance. Patients can regain strength and motor function through exercises and games.

Rehabilitation therapy is defined by three main forms: passive, active, and active-assisted therapy. Passive rehabilitation therapy is when the only force applied for the patient to complete their exercise is external, meaning the patient inputs no effort. This allows patients with more limited functions to begin their rehabilitation therapy. Active therapy is when no external force is used to complete an exercise, and patients need to exert all their effort to complete the exercise. This step is crucial for the patient, as active exercises strengthen the hand muscles and solidify their neuroplasticity. Active-assisted therapy combines both forms, providing the patient with an opportunity to strengthen their muscles much more efficiently. Rehabilitation requires all three phases to return the patient to proper motor function. By performing all three stages, patients can connect their muscle gain to their neural paths, satisfying the conditions for neural plasticity. As the population grows, the demand for rehabilitation therapy and assistive devices will increase. Currently, the demand for physical therapists outpaces the number of trained professionals available [13,14]. Furthermore, rehabilitation therapy is not limited to only post-stroke victims but can be used for any injury-causing muscular atrophy or loss of strength. While other injuries continue to occupy physical therapists, the limited number of sessions is even more tapered [14]. 

Rehabilitation therapy progress is frequently measured using scales to determine the severity of motor function loss, the capabilities of the affected limb, and the change in functional exercises. Judging progress for rehabilitation has no global uniformity, but generalized examinations have been created by professional medical associations. First, a patient’s stroke severity is determined by the Chedoke-McMaster Stroke Assessment (CMSA) [15]. The CMSA helps categorize patients’ impairment levels and can be further adapted using the Barthel Index for increased levels of vascular brain damage [16]. With the severity now categorized, the second step is using a primary assessment for motor function recovery, the Fugl-Myer Assessment (FMA). This assessment is a 226-point, Likert-type scale developed to measure a patient’s recovery from a stroke by dividing the process into five domains: motor function, sensory function, balance, joint range of motion (ROM), and joint pain [17]. FMA scores are used to evaluate the efficiency of rehabilitation devices with their exercises encompassing the main indicators for rehabilitation progress.

Furthermore, Krakauer studied several stroke patients and their progress through their rehabilitation and noted that patients would not meet a benchmark of eleven by the fourth week of rehabilitation. They are only 6% likely to recover full motor function by six months [18]. Since the score is based on severity and can be adjusted to target specific limbs, the FMA cannot fully characterize patients’ needs. This issue is curtailed with the help of medical professionals who gauge how to adjust the progress using previous experience and medical knowledge.

The need for rehabilitative or assistive robotics is high. The market for powered rehabilitation devices in North America in 2020 is approximately USD 500 million [19], which is not surprising given the high-risk Americans have of heart disease and other bodily injuries. Furthermore, these devices estimated a compounded annual growth rate (CAGR) of 22.6% annually until 2026 [19,20]. Although North America may currently have the largest market, the aging populations of Asia are expected to drive demand much higher.

Powered prostheses have been studied since the early 1900s and have since evolved into more robust machines [21]. Although not portable, these powered devices used pneumatic actuation to simulate grasping motions. Quickly, components became much smaller and could incorporate elaborate control algorithms, such as PID control [21]. In addition, the first prosthetic to incorporate multidisciplinary motion and myoelectric control was developed in 1970 and is known as the SVEN-Hand [22]. With the expanding developments of prosthetics, development branched into powered rehabilitation.

Since their initial designs, these devices have advanced, employing more sensing capabilities, actuation devices, and overall weight [23]. Powered rehabilitation devices provide rehabilitation therapy with the same efficiency as regular rehabilitation [23,24]. Another use for assistive robotics’ is with the elderly or in nursing care. With social isolation becoming more common, especially due to increased remote work, socially assistive robots have helped elderly patients stay active, are used to monitor their health, and assist in daytime organization [25]. Other devices deployed include social facilitation, health screening, monitoring, and emergency alerts [25]. Although the devices cannot actuate or provide any physical assistance, patients can still become more accustomed to automated interfaces [26]. Direct evidence of neural plasticity resulting from rehabilitation therapy is clear and further expands on how robotic rehabilitation therapy can achieve the same goals as in-person rehabilitation [27,28]. The FMA scores for patients undergoing in-person rehabilitation therapies are almost indistinguishable from those undergoing rehabilitation remotely or using a powered device. Since rehabilitation is accomplished using either method, powered devices are a proven solution to rehabilitation therapy demand [29,30].

In this paper, the discussion is focused on the design, actuation mechanisms, control, and applications for hand rehabilitation devices. The study’s primary purpose is to create a basis for an at-home rehabilitation device capable of conducting passive, active, and active-assisted exercises with a neurological feedback system. This device would help post-stroke patients rehabilitate their hands for improved strength and motor function. The organization of the manuscript is shown in Figure 1. 

## 2. Journal Selection Method

The literature reviewed in this paper was limited to rehabilitation devices for the hand. Devices were researched using six databases such as Google Scholar, National Library of Medicine, IEEE Xplore, MDPI, PubMed, and Scopus. Only papers written in English are included while excluding devices with little technical information to narrow the search. Keywords such as hand rehabilitation device, active, passive, and active-assisted rehabilitation therapy, hand exoskeleton, and hand orthosis were used to compile these powered devices into a list. These devices were further categorized based on their mechanical design, actuation mechanisms, control methods, and interfaces.

To properly conduct a literature review, criteria for the rehabilitation device must be defined. The papers retrieved were further filtered to devices for upper limb rehabilitation. Within this category, hand, wrist, elbow, and complete arm devices were discovered. Devices that did not primarily rehabilitate the hand and fingers were removed from the search. Moreover, only rehab devices categorized as orthoses, exoskeletons, or end-effector devices were considered for this search. Figure 2 shows the systematic approach with inclusion and exclusion criteria for the selected studies. Active exercises require no power, so devices providing solely active rehabilitation are primarily orthoses. Although some provide resistive training, orthoses still cannot provide passive rehabilitation therapy but provide compact solutions for integral rehabilitation therapy. Devices capable of passive exercises can also be used for active exercises and are further classified by their actuation method.

After producing 326 papers meeting the criteria, 113 were removed since they did not apply to hand rehabilitation or did not incorporate passive exercises. Of 213 papers, 36 were crucial for this study. Although active rehabilitation is integral, medical devices have advanced far enough to incorporate both forms of rehabilitation. Furthermore, the study focused on devices that have been trialed by patients or are available commercially.

## 3. Hand Kinematics

In the human hand, all fingers, excluding the thumb, contain three phalanges, known as the distal phalange, middle phalange, and the proximal phalange. The thumb contains only two bones and no middle phalange. To design an exoskeleton rehabilitation device for the hand, the device’s link lengths and degrees of freedom (DoFs) should correspond to the anatomical configuration of the finger joints. Excluding the thumb, the phalanges comprise three joints, the metacarpophalangeal (MCP) joint, the proximal interphalangeal (PIP) joint, and the distal interphalangeal (DIP) joint. The thumb comprises only two joints, MCP and DIP joints. The MCP joint of the fingers has 2 degrees of freedom. It provides abduction and adduction with flexion and extension motions, whereas the PIP and DIP joints only have one degree of freedom and solely provide flexion and extension motions. The axes of rotation $(Z_{i}$) of each joint are shown in Figure 3. To find the forward kinematics of the index, middle, ring, and little fingers, the modified Denavit-Hartenberg (DH) parameters corresponding to a link-frame attachment are shown in Table 1, and that for the thumb is shown in Table 2. 

Where,

Link Length ($a_{i}$) is the length measured ($L_{i})$ along with *X_i_*, from axis *Z_i_* to axis *Z_i+_*_1_;

Link Twist ($\alpha_{i}$) is the angle measured about *X_i_*, from axis *Z_i_* to axis *Z_i+_*_1_;

Link Offset ($d_{i}$) is the distance measured along the axis *Z_i_*; from *X_i−_*_1_
*to X_i_*, and

Joint Angle ($\theta_{i}$) is the angle measured ($q_{i}$) about *Z_i_*, from *X_i−_*_1_
*to X_i_*

These DH parameters are used to get the homogeneous transfer matrix, which represents the positions and orientations of the reference frame with respect to the fixed reference frame. It is assumed that the fixed reference frame $\left\{0\right\}$ is located at the same point as the first reference frame $\left\{1\right\}$ As shown in Figure 3. The general form of a link transformation matrix that relates to frame $\left\{i\right\}$ relative to the frame $\left\{i-1\right\}$ ca be expressed as [31]: (1)Tii−1=Rii−13×3Pii−13×101×31
where, Rii−1 is the rotation matrix that describes the frame $\left\{i\right\}$ relative to frame $\left\{i-1\right\}$ and can be expressed as:(2)Rii−1=cosθi−sinθi0sinθicosαi−1cosθicosαi−1−sinαi−1sinθisinαi−1cosθisinαi−1cosαi−1
and Pii−1  is the vector that locates the origin of the frame $\left\{i\right\}$ relative to frame $\left\{i-1\right\}$ and can be expressed as:(3)Pii−1=ai−1−sαi−1dicαi−1diT

Using Equations (1)–(3), the individual homogeneous transfer matrices that relate two successive frames (Figure 3) of the index, middle, ring, and little fingers and using the modified DH parameters from Table 1 can be found using Equations (4)–(8):(4)T10=cosθ1−sinθ100sinθ1cosθ10000100001
(5)T21=cosθ2−sinθ20000−10sinθ2cosθ2000001
(6)T32=cosθ3−sinθ30L1sinθ3cosθ30000000001
(7)T43=cosθ4−sinθ40L2sinθ4cosθ40000000001
(8)Tf4=100L3010000100001

Similarly, using the Equations (1)–(3), the individual homogeneous transfer matrix that relates two successive frames (Figure 3) of the thumb will use the modified DH parameters of Table 2 and can be found using Equations (9)–(12):(9)T10=cosθ1−sinθ100sinθ1cosθ10000100001
(10)T21=cosθ2−sinθ20000−10sinθ2cosθ2000001
(11)T32=cosθ3−sinθ30L1sinθ3cosθ30000000001
(12)Tf3=100L2010000100001

Thus, the homogenous transformation matrix that relates the fingertip frame $\left\{f\right\}$ to the fixed frame $\left\{0\right\}$ can be obtained by multiplying individual transformation matrices. Equations (9)–(12) represent the forward kinematic equations of the index, middle, ring, and little fingers, and that for the thumb finger is shown in Equations (13) and (14):(13)Tf0=T10·T21T43·Tf4
(14)Tf0=T10·T21·Tf3

The studies of dynamics discuss the rigid body’s motions and the forces and torque that cause them. The dynamic modeling of the finger using the iterative Newton-Euler formation can be expressed by the well-known rigid body’s dynamic equation as follows [32]:(15)τ=Mθθ¨+Vθ,θ˙+Gθ
where Mθ is the n×n mass and inertia matrix of the fingers (*n* = 3 for the thumb, but *n* = 4 for the other fingers), Vθ,θ˙ is an n×1 vector of centrifugal and Coriolis terms, and Gθ is an n×1 vector of gravity terms. Anatomic data of human fingers such as mass, inertia, length, and center of gravity can be found in [33]. After including the finger’s resistance force due to impairment or spasticity, the dynamic equation of the human finger (15) can be rewritten as follows:(16)τ=Mθθ¨+Vθ,θ˙+Gθ+JθTF
where Jθ∈ℝ6×n is the Jacobian matrix that relates the fingertip’s linear and angular velocities with the joint’s (MCP/DIP/PIP) velocity; and
(17)F=fxfyfzτxyτyzτzxT∈ℝ6×1

*F* is the applied force and torque vector at the fingertip.

The most important design requirements are the assistive force and range of motion needed for each joint. Several healthy patients were tested with 11 activities of daily living (ADLs) to determine healthy ROM of the MCP, PIP, and DIP joints [34,35]. From the measured values, the healthy ROM for the joints is shown in Table 3. Once the kinematics of the hand is resolved, the next step in designing an assistive device is to find the limits of the patients using the device. In the design of the Gloreha device, Borboni et al. tested the capabilities of four patients in need of hand rehabilitation [36]. To determine the flexion and extension forces exerted by patients within the limits of their ROM and pain tolerance, they adapted the Wheatstone bridge concept composed of a metallic rod connected to four strain gauges. Here, a physiotherapist would aid the patients with a rehabilitation exercise and record the forces measured through the gauges. The results of the test for a pathological hand for each patient are shown in Table 4. 

## 4. Hand Rehabilitation Devices

Hand rehabilitation devices (HRDs) can be characterized as devices or braces designed for any stage of rehabilitation therapy for the hand. Rehabilitation therapy for the hand has been defined in set stages of exercises as outlined by the University of Kentucky’s Healthcare department (UKHC) [37]. Here, UKHC defines flexion and extension of the MCP, PIP, and DIP joints at different angles from the beginning to the final stages of hand rehabilitation.

HRDs are classified as orthoses, exoskeletons, or end-effector devices. Each form provides crucial benefits to the rehabilitation process. Orthoses provide essential support and operate like most hand braces. Exoskeletons and end-effector devices add a necessary component to rehabilitation therapy: passive exercises. These powered devices offer crucial support and actuation of the phalanges using varying actuation mechanisms, including pneumatic and linkage-based systems, and control methods, including PID (proportional integral derivative) and admittance control. Several rehabilitation devices for the hand employ solely active or passive exercises. Since active rehabilitation is accomplished without external force, devices providing solely active rehabilitation therapy are primarily orthoses.

Exoskeletons and End-effector devices provide patients with both active and passive rehabilitation, combining multiple solutions into one device. Incorporating actuation into rehabilitation devices increases motor recovery and rehabilitation compared to standard in-person rehabilitation with a physical therapist [38,39]. While no significant advantages or disadvantages in rehabilitation progress are clear between exoskeletons and end-effector devices that provide active and passive rehabilitation, the kinematic model of flexion and extension of the joints are the same [40,41]. The mechanical designs of an exoskeleton and end-effector device is shown in Figure 4. 

### 4.1. Exoskeleton Devices

Exoskeletons are devices designed to fit over the target area in a similar way to an orthosis, in that exoskeletons provide safety and support; however, exoskeletons have additional components that provide powered functions to the end-user. Exoskeletons are designed as wearable electromechanical devices that enhance the physical performance of the wearer [44]. For hand rehabilitation, patients with more severe injuries require additional force to begin the rehabilitation. The actuated devices reviewed in this paper are listed in Table 5. Heavily cited exoskeletons reviewed in this paper are shown in Figure 5.

Exoskeletons provide solutions unique to their design. Since exoskeletons encompass each individual finger, actuation over individual joints while also providing support is crucial for patients with advanced injuries. Furthermore, exoskeletons provide portable solutions necessary for elderly or remote patients. Major drawbacks with exoskeletons include complex control algorithms and the adjustability of the device. Since an exoskeleton will actuate each individual appendage over each joint, controlling positioning requires complex strategies. Moreover, exoskeletons must be tailored to fit a patient’s hand. With end-effector devices, the mechanical designs are focused on end-point control and can adjust to varying hand dimensions.

A mathematical model for a hybrid-driven exoskeleton [71] for a hand is provided in Figure 6. The device is described as hybrid-driven as the device is actuated using active tendon-driven actuation and passive flexible-hinge joints. The following equations were derived from Meng’s development of the Variable Stiffness Fixed Hinge (VSFH) device [71]. Using Figure 6a,b, the kinematics of a single finger is projected onto a two-dimensional xy-plane. Further simplifying the model, Meng shows how a finger can be modeled similarly to a cantilever beam using Bernouilli’s equations. These equations are replicated in Equations (18)–(22). Here, the distances between each joint, also known as the proximal, intermediate, and distal phalange, are denoted as $L_{i},\;l_{i}$. Each joint will require an extension of the beam, as denoted by $\Delta L_{i}$. If $\varphi_{i}$ represents the angle of each joint, h represents the distance of the exoskeleton to the finger, and t represents the thickness of the finger. Then, the change in the length of the beam is given by Equation (18). Subjecting the exoskeleton modeled as a cantilever beam to forces $F_{0}$ and moment $M_{0}$. The tip’s slope and end-point angle are given by $\phi_{i}$ and $\theta_{i}$ as shown by Figure 7. Further, with Young’s modulus E and the moment of intertia of the beam I, the change in the joint’s length *l*, and the deflections at the tip a,b  can be calculated using Equations (19)–(22).



(18)
$$\Delta L_{i}=\left(h+\frac{t}{2}\right)$$


(19)
$$k=\frac{d\theta }{ds}=\frac{\frac{d^{2}y}{dx^{2}}}{{\left[1+{\left(\frac{dy}{dx}\right)}^{2}\right]}^{\frac{1}{2}}}=\frac{M}{EI}$$


(20)
$$L=\int^{\theta }\frac{d\theta }{k}$$


(21)
$$a=\int^{\theta }\frac{\cos\left(\theta \right)}{k}$$


(22)
$$b=\int^{\theta }\frac{\sin\left(\theta \right)}{k}$$



### 4.2. End-Effector Devices

End-effector devices differ from exoskeletons because the actuation over the joints is based on end-point control, or control over the DIP joint, rather than the entire finger [28,75]. Typically, these devices are stationary since the hand would need to be fixed for the DIP joint to be actuated with proper position control. Figure 8 shows all the end-effector HRDs reviewed in this paper. Portability does provide helpful, remote rehabilitation for patients but can limit the device in terms of possible rehabilitation and progress tracking. Aubin et. al. notes how end-effector devices can adjust to the patient’s right or left hand and dimensions, providing a singular solution for varying injury severity and hand size [76]. 

One main advantage of end-effector devices is the high level of control and feedback. Using linear rails and a control system that reads and displays neurological responses during the exercises, the Amadeo can actuate and rehabilitate the hand for patients of all ages and severity [46]. The Haptic Knob can provide accurate control using force feedback and impedance control over the sphere [43]. Additionally, end-effector devices can employ more sensing components since portability is no longer a design aspect [76].

Clinical trials have already been completed for the reviewed exoskeletons and end-effector devices. Amadeo, Gloreha, HEXORR, and My-HERO reported an increase of at least 7 points on the FMA scale, outperforming control groups of standard rehabilitation therapy [46,60,77,78,79,80]. Smaller, more portable devices such as VAEDA, PneuGlove, ReHand, and the device designed by Tong et al. improved the FMA scores by 2–4 points [66,67,68,81]. Most devices were able to improve FMA scores by 2–4 points [47,82,83,84]. Both end-effector and exoskeletons improve the FMA scores by varying scales, indicating no correlation between the powered devices’ structures and overall effectiveness.

End-effector devices provide key advantages over exoskeletons. End-effector devices incorporate much simpler control mechanisms without sacrificing overall rehabilitation progress. Since the FMA scores aren’t significantly different, end-effector devices provide high levels of control and safety without complicated mechanical designs. Another advantage is the increased amount of user feedback. Devices such as the Amadeo can display neurological profiles of a patient’s hand while the exercises are conducted. However, end-effector devices are not able to constrain each individual appendage. Since the end-point control only actuates from the end of a finger, the risk for malalignment increases [85].

### 4.3. Hand Orthoses 

Devices categorized as orthoses are devices created to protect and support the hand [86,87]. Table 6 lists the orthoses reviewed in this paper, and Figure 9 shows the orthoses reviewed. These devices all provide active therapy, but only one provides resistive training. Since these devices do not typically include any powered components, the therapy is restricted to patient input. This limitation prevents patients with more severe injuries from receiving proper treatment [87].

Some orthoses employ resistive therapies because they provide strengthening exercises without any powered components. SCRIPT and Saebo orthoses employ spring cords to provide resistive training in their device [88,89]. Resistive exercises are a part of late-stage hand rehabilitation since their primary purpose is to strengthen the muscles rather than reintroduce proper movement and placement [37]. 

**Figure 9 micromachines-13-01033-f009:**
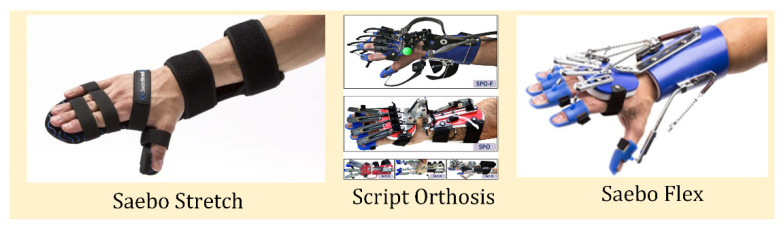
All orthoses reviewed in this paper are shown [88,90,91]. These devices provide the basic support and alignment necessary for rehabilitation.

**Table 6 micromachines-13-01033-t006:** Reviewed Orthoses.

Name	Exercise Types	Sensors	Mechanism	Weight
**Saebo Stretch** [90]	A	-	Resistive plastic	-
**Saebo Flex** [91]	A, R	-	Resistive spring	-
**SCRIPT** [88]	A, R	Force, torque, and inertial sensors, adjustable potentiometer	Resistive spring	0.65 kg

A = active, R = resistive.

## 5. Actuation Mechanisms of Hand Rehabilitation Devices

Actuation mechanisms describe mechanisms used in devices to translate power transmitted from actuators in exoskeleton or end-effector devices. Mechanisms employed by devices reviewed in this paper are categorized into four main classifications: pneumatic actuation, linkage-based actuation, cable-driven systems, and gear-motor actuation. Pneumatic actuators require sealed tubes or conduits to actuate the devices, whereas linkage-based actuation requires only direct connection from the actuator to the finger. Cable-driven systems translate actuation using cables connected throughout the finger. Gearmotor actuation employs gearmotors and gear trains primarily for increased torque transmission for the hand. Table 7 presents the actuated devices reviewed in this paper along with the actuators and actuation mechanisms employed. 

### 5.1. Pneumatic Actuation

Typically, pneumatic devices are powered by air pistons, using pneumatic tubes or pathways to translate the actuation for total flexion of the MCP, DIP, and PIP joints. Pneumatic actuation provides ample torque and control for rehabilitation. Pneumatic devices include the Rutgers Master II and the Power Assist Glove [47,53]. A key advantage of pneumatic devices is their weight-to-torque ratio. The SymbiHand only adds 241 g of mass to the hand, and the Power Assist glove only adds 170 g [36,52]. However, even though the SymbiHand is modeled after pneumatic devices, it employs an electrohydraulic system. Moreover, the Gloreha glove actuates the hand using air-filled bladders rather than being directly actuated from a pump. These unique solutions have been cited for having positive effects on their patients.

The main drawback of pneumatic devices is the space required. Pneumatic and hydraulic systems are dependent on multiple components, including the compressor, separate chambers, pumps, and additional power sources. Low-weight options are available, but pneumatic devices may impede exoskeleton designs with a greater need for telerehabilitation and precise control [92]. Furthermore, compressed air operates well inside the closed loop, but the actuation conducted by the devices is often slow in response. Increasing rotational speed requires higher-cost solutions for better torque and control [93]. 

### 5.2. Linkage-Based Actuation 

Linkage-based actuation is a popular choice for remote rehabilitation. Using links, either 3D printed or machined, linear actuators and varying motors translate the powered actuation. Devices that employ this method include the Hand of Hope, CyberGrasp, and HEXORR. These devices employ linkages to provide planar movement for each finger, securing the range of motion and preventing patients from flexing or extending out of the plane [92]. Linkage-based actuation is versatile and can be implemented with a range of control methods, whereas pneumatic actuation is limited to pneumatic controllers. By combining the actuators with a rigid exoskeleton, the flexion and extension motion can be achieved with simple and often 3D printed linkages [92,94]. 

Linkage-based actuation does have a disadvantage. Motors and the necessary translational linkages can add weight to the system, even though the motion is simple [94]. To provide a compact design, more expensive actuators can be employed at the cost of other sensing components that could be integral to the patient’s rehabilitation process, as seen by the Exo Glove by Festo and Cyber Grasp [49,58]. Furthermore, the addition of linkages complicates the kinematics of the exoskeleton. With other forms of actuation mechanisms, the kinematics does not alter as much as linkage-based systems in terms of the hand’s dimensions. With pneumatic actuation, the kinematics would not change; however, with linkage-based systems, each link’s mass, inertia, and center of gravity change for each hand. 

### 5.3. Cable-Driven

Cable-driven exoskeletons employ servo or rotary motors to provide enough torque to rehabilitate the hand. Systems such as the WearME, SEM Glove, and the glove designed by Delph et al. use cables for low-weight solutions. Implementing cable-driven systems allows the device to shift the weight of the actuation components away from the hand [54,57,74]. Another advantage of cable-driven systems is the soft-exoskeletons used to frame the device. Each hand has different shapes and sizes, and soft, cable-driven devices provide an adaptable design that can adjust to the patient’s dimensions [24,95].

A major disadvantage of cable-driven systems is the loss and control issues. Since the system depends on translating the actuation via cables, the actuation experiences transmission losses as the device work through the exercises [95]. Cable-driven systems are more prone to friction losses since that cable must rely on a spool to release and retract the driving cable. As a result, there is a problem with control. As pictured in Figure 10, cable-driven exoskeletons like the WearME device can become overburdened by the mechanisms needed to actuate and control the device fully. Because the cable’s position can vary with adjustable designs, multiple control methods must be used in addition to position control. The Delph’s hand employs three control methods to provide accurate control: force, position, and sEMG [74].

### 5.4. Gear-Motor Actuation

Gear-motor actuation mechanisms include any translation of power and motion using a combination of gear motors or gear trains. Devices such as the Exo-K’ab use gear motors for precise control over the joints. This mechanical control over the rotation is beneficial to actuation design since it provides safe and accurate control over 360°. Although the MCP, PIP, and DIP joints do not require 360° motion, the fluid motion provides beneficial rehabilitation.

Gear-motor actuation falls short in terms of cost and implementation. Gear motor actuation is often an expensive approach to actuation. With a human hand, several gears and gear trains would be employed to provide actuation. Moreover, designs using gear-motor actuation run into similar weight issues experienced by linkage-based actuation. Both forms require heavy components to provide the safety and security pneumatic actuation can provide. 

## 6. Therapy Strategies of the Hand Rehabilitation Devices

Current forms of robotic therapies aid beyond normal passive and active movements. Several devices incorporate passive-assistance in addition to fully passive motion to aid patients with more severe injuries. Devices such as Amadeo and CyberGrasp also incorporate games to better interact with the patient using the device. These games not only help engage patients with repetitive tasks, but also provide challenge-based goals to help progress rehabilitation. In this section, the only devices compared are those with clinical trials with proven FMA score improvements.

### 6.1. Assistive Strategy

Assistive strategies provide assistive actuation based on user intention and end goals. Devices that implement this level of control have set goals for the patient to accomplish, such as certain hand signs or grasping and pinching motions. When users plan on closing their hand but lack the strength to drive their hand, the assistive control methods then aid the patient in completing the exercise based on need [60,85]. This control strategy differs from admittance control as it can be used for patients that struggle to activate the low-level force sensors. Another circumvention of the assistance needed is using voice activation, like the VAEDA [68]. 

### 6.2. Haptic Strategy

Haptic strategies are an emerging strategy for hand rehabilitation. Haptic control is defined as a method for the patient to use a virtual system to replicate sensation or motion. The Hand of Hope and Rutgers Master II were trialed with patients to test their implementation of haptic control. The two devices performed well and were able to improve the FMA scores of the patients [83,96]. Additionally, the Amadeo and CyberGrasp incorporate VR games and exercises to provide haptic therapy [49,97].

### 6.3. Challenge-Based Strategy

Challenge-based strategies are implemented by requiring the end-user to accomplish an intuitive goal. This can be accomplished by having patients participate in exercises or games to encourage the patient to solve the challenge. Challenge-based strategies work well since the gamification of exercises increases patient involvement and practicing challenges improves motor function faster than simple steps [98]. Some devices that use challenge-based control include Amadeo and Gloreha [79,80].

### 6.4. Coaching Strategy

Coaching is a crucial strategy and is necessary for other forms rehabilitation. Coaching refers to the rehabilitation device not providing assistance or resistance, but simply measuring motion and user intention using the device’s sensors and providing feedback to the user [99]. Since conventional physical therapy has a motivational aspect, coaching can help users track their progress while building independence during the therapy. The coaching control method can be seen with the Amadeo or ReHand. The Amadeo provides a user interface to view a patient’s rehabilitation progress for both the physical therapists and the patient, whereas the ReHand showed trajectory and force changes during the trial [67,80].

### 6.5. Telerehabilitation

Rehabilitation has now incorporated videogames and exercises to engage patients further in the rehabilitation progress [100]. Allowing patients to continue their rehabilitation on their own allows for faster recovery time and lower medical costs. Telerehabilitation offers the perfect balance of at-home solutions with the care of a professional. Telerehabilitation effectively provides the necessary exercise for patients [101,102]. Furthermore, in combination with VR or exercise games, telerehabilitation yielded high satisfaction from the patients who use the system [103,104].

The main obstacle facing telerehabilitation is accurate position control. Since the rehab devices already employ highly accurate control features, designs of the SCRIPT or HEXORR, some devices created for telerehabilitation are completely safe [105,106]. Most patients who have opted for telerehabilitation instead of face-to-face clinics have reported higher satisfaction levels [107].

Table 8 shows an accurate comparison of devices with clinical trials and the rehabilitation strategies employed. Devices that employed simple passive, active, and assistive rehabilitation still improve the FMA scores of patients. However, the largest improvements are seen with devices with more advanced strategies, such as challenge-based or coaching rehabilitation, like the Amadeo and Gloreha [79,80].

## 7. Control Methods of the Hand Rehabilitation Devices

Control methods are applied to manipulate HRDs to provide different forms of rehabilitation therapy, including passive, active, and active-assisted therapy. Various controllers are employed to achieve different goals during the rehabilitation exercises, such as position control, force control, or torque control [108,109,110,111,112,113]. These controllers operate using pre-determined inputs to achieve desired outputs such as torque, end-effector force, or position or trajectory values. In passive rehabilitation mode, the exercises are conducted by an input command to the controller [109,113,114,115,116,117]. On the other hand, in active-assisted or resistive therapeutic approaches, the patient’s exerted force is measured, commonly with surface EMG signals or force sensors, and is used as a control input to the robot controller [109,110,118,119,120,121,122]. Research shows that most HRDs use model-free PID (proportional integral derivative) control [31,112,116,123]; others use model-based nonlinear controls [113,114,124,125], admittance control [122,126], and impedance control [127,128]. PID [112,116,123] and model-based nonlinear controllers, such as computed torque control (CTC) [113,114] and sliding mode control [114,124], are used mainly to maneuver the HRDs to provide passive rehabilitation therapy [31]. On the other hand, admittance and impedance controllers are used to control the HRDs motion to provide active, active-assistance, and resistive therapy [120,122,128].

The earliest control method for prosthetics was simple motor control over the actuator [21]. However, the human hand consists of 27 degrees of freedom, requiring much more robust control and actuation. With the advancements in manufacturing and computation, control strategies could now incorporate adaptive control using PID control [99,129]. These powered devices, while advanced initially, didn’t account for variation in motion. Once myoelectric control was integrated into powered devices, control strategies needed to evolve to incorporate pattern recognition strategies further.

### 7.1. Position Control with PID

The PID control technique is the most widely used control technique for industrial applications [31]. It is simple in design and efficient in computation. Moreover, it is considered a robust control technique. PID control is often used to prove passive rehabilitation therapy. The general layout of the PID control approach is depicted in Figure 10. The joint torque commands of the HRD can be expressed by the following equation:(23)τ=KPθd−θ+KVθ˙d−θ˙+KI∫θd−θdt
where θd,θ∈ℝn are the vectors of desired and measured joint angles respectively, θ˙d, θ˙∈ℝn are the vectors of desired and measured joint velocities respectively, KP, KV, and KI are the diagonal positive definite gain matrices, and τ∈ℝn is the generalized torques vector. Let the error vector E and its derivative be:(24)E=θd−θ; E˙=θ˙d− θ˙

Therefore, this Equation (23) can be re-formulated as an error equation:(25)$$\tau =K_{P}E+K_{V}\;\dot{E}+K_{I}\int \mathrm{Edt}$$

The relation (25) is decoupled, therefore individual torque command for each joint would be as follows.
(26)$$\tau_{i}=K_{P_{i}}e_{i}+K_{V_{i}}{\dot{e}}_{i}+K_{I_{i}}\int e_{i}d$$
where $e_{i}=\theta_{d_{i}\;}-\theta_{i}\cdots \left(i=1,\;2,\cdots n\right)$; $\theta_{i},{\;\theta }_{d_{i}\;}$ are the measured and desired trajectory for joint i respectively, and
$$\dot{E}={\left[\begin{array}{cccc} {\dot{e}}_{1} & {\dot{e}}_{2} & \cdots & {\dot{e}}_{n} \end{array}\right]}^{T},$$
$$\theta_{d}={\left[\begin{array}{cccc} \theta_{d_{1}} & \theta_{d_{2}} & \cdots & \theta_{d_{n}} \end{array}\right]}^{T},$$
$$\theta ={\left[\begin{array}{cccc} \theta_{1} & \theta_{2} & \cdots & \theta_{n} \end{array}\right]}^{T};$$
$$K_{P}=\mathrm{diag}{\left[\begin{array}{cccc} K_{P_{1}} & K_{P_{2}} & \cdots & K_{P_{n}} \end{array}\right]}^{T},\;$$
$$K_{V}=\mathrm{diag}{\left[\begin{array}{cccc} K_{V_{1}} & K_{V_{2}} & \cdots & K_{V_{n}} \end{array}\right]}^{T},\;\mathrm{and}$$
$$K_{I}=\mathrm{diag}{\left[\begin{array}{cccc} K_{I_{1}} & K_{I_{2}} & \cdots & K_{I_{n}} \end{array}\right]}^{T}$$

Shown in Figure 11 are the position trajectory results of one passive exercise routine conducted by a patient using the ExoK’ab device. The blue and yellow lines represent the device’s desired trajectory for each phalange, and the red and green lines represent the actual movement of the patient’s finger. Here, Sandoval compares the simulated and experimental movements of a patient when conducting a passive exercise. 

### 7.2. Model-Based Computed Torque Control

To realize better tracking performance of an HRD, the dynamic model of the HRD (as well as the dynamic model of the human fingers) needs to be included in the control law. Therefore, the nonlinear CTC technique is often used for this. Its control law includes both the finger and HRD’s dynamic model. The dynamic behavior of an HRD can be expressed by the well-known rigid body dynamic equation: (27)Mθθ¨+Vθ,θ˙+Gθ+Fθ,θ˙=τ
where θ∈ℝn is the joint variables vector, τ  is the generalized torque vector, Mθ∈ℝn×n   is the inertia matrix, Vθ,θ˙∈ℝn is the Coriolis/centrifugal vector, Gθ∈ℝn is the gravity vector, and Fθ,θ˙∈ℝn is the friction vector. Note that the friction vector is modeled as nonlinear Coulomb friction and can be expressed as:(28)τfriction=Fθ,θ˙=c.sgnθ˙
where c is the Coulomb-friction constant. Equation (27) can be written as:(29)θ¨=−M−1θVθ,θ˙+Gθ+Fθ,θ˙+M−1θτ

$M^{-1}\left(\theta \right)$ always existed since $M\left(\theta \right)$ is symmetrical and positive definite.

The layout of the modified computed torque control technique is depicted in Figure 12. The control torque in Figure 12 can be written as:(30)$$\tau =M\left(\theta \right)\left[{\ddot{\theta }}_{d}+K_{V}\left({\dot{\theta }}_{d}-\dot{\theta }\right)+K_{p}\left(\theta_{d}-\theta \right)+K_{i}\int \left(\theta_{d}-\theta \right)\mathrm{dt}\right]\;+V\left(\theta ,\dot{\theta }\right)+G\left(\theta \right)+F\left(\theta ,\dot{\theta }\right)$$

From relations (27) to (30), we may write:(31)$$\ddot{\theta }={\ddot{\theta }}_{d}+K_{V}\left({\dot{\theta }}_{d}-\dot{\theta }\right)+K_{p}\left(\theta_{d}-\theta \right)+K_{i}\int \left(\theta_{d}-\theta \right)\mathrm{dt}$$
where ${\;\theta }_{d},{\dot{\theta }}_{d},\;\mathrm{and}\;{\ddot{\theta }}_{d}$ are the desired position, velocity, and acceleration, respectively, and $K_{p}$, $K_{V}$, and $K_{i}$ are diagonal positive definite matrices. Let the error vector E and its derivative be:(32)$$E=\theta_{d}-\theta ;\dot{E}={\dot{\theta }}_{d}-\dot{\theta },\;\ddot{E}={\ddot{\theta }}_{d}-\ddot{\theta }$$

Therefore, Equation (31) can be rewritten in the following form:(33)$$\ddot{E}+K_{V}\dot{E}+K_{p}E+K_{i}\int \mathrm{Edt}=0$$
where the control gains $K_{p}$, $K_{V}$, and $K_{i}$ are positive definite matrices. Therefore, a proper choice of these matrices ensures the stability of the system.

### 7.3. Admittance Control

In active, active-assistive, resistive rehabilitation mode, the HRDs assist/resist user finger motion/exercise based on the human-user interactive force measured from the force sensor or skin surface EMG signals. To do this, the controller partially modifies the desired trajectory/exercise of an HRD in relation to the force sensor/EMG input, as expressed by Equation (34).
(34)$$q_{a}={\;q}_{d}+\;\tau \;\left(\frac{1}{K+C_{s}}\right)$$
where, $q_{a}$ is the nx1 vector of the new desired trajectory defined by the admittance, $q_{d}$ is the nx1 vector with the desired initial trajectory from the trajectory planner. τ is the joint torques that include torques due to human-robot interactive force/EMG signals. K and $C_{s}$ are the gain matrices corresponding to a spring and damper constant, respectively. Adjusting these constants provide higher or lower resistance to subjects’ movements using force or EMG inputs. Since the admittance requires direct user intention, patients with more severe injuries may find it difficult to use [68].

### 7.4. Impedance Control

Impedance control can be demonstrated using an impedance model provided by Ibarra’s review of impedance control [130]. An impedance controller using force feedback uses Equation (35) reproduced from Wang’s calculations [127].
(35)$$F=m_{d}\left(\ddot{\theta }-{\ddot{\theta }}_{d}\right)+b_{d}\left(\dot{\theta }-{\dot{\theta }}_{d}\right)+k_{d}\left(\theta -\theta_{d}\right)$$
where $m_{d},{\;b}_{d},{\;\mathrm{and}\;k}_{d}$ represent the inertial, damping, and stiffness matrices, respectively, *F* is the force vector, $\ddot{\theta }$ and ${\ddot{\theta }}_{d}$ represent measured and desired angular acceleration, $\dot{\theta }$ and ${\dot{\theta }}_{d}$ represent measured and desired angular velocity, and θ and $\theta_{d}$ represent mesured and desired joint angle, respectively. The input values are sent to a controller using a dynamic transformation using the three variables, angular acceleration, velocity, and position. A control schematic of an impedance-based control that employs force or EMG feedback is shown in Figure 13.

Where $K_{fp}$ is the coefficient of force and position, $\;K_{fv}$ is the coefficient of force and velocity, T is the kinematic equation for the device, and J is the Jacobian matrix. Once the output angular speed is fed back into the speed control loop, angular speed control is implemented, allowing for integration with the position control loop. Since rehabilitation devices are actuated slowly for the safety of the patient, the angular acceleration is negligible. Once the force feedback sensors detect interaction from the patient, the device’s end signal can be inverted to obtain a control signal to assist the patient. This strategy is often used as a complement to admittance controls since some sensors, such as torque and position sensors, can double as sensors for both [126]. Since the method is not very complex, impedance control can be inaccurate due to the opposing motions. Admittance control can overcome this issue since the motion is not resisting user intention but can encounter errors when the motion is more dynamic rather than linear movements [131].

As with all medical devices, precise feedback control is critical. Without proper control or feedback from the device, the patient could experience tremendous pain/fatigue and potentially cause more damage to their hand [24,93]. The control must be precise for remote rehabilitation and provide real-time haptic feedback to the patient and the clinician assisting virtually. Several control strategies have been proven to provide safe and accurate control with proper sensors. Both end-effector and exoskeleton devices can employ several control strategies using varying combinations of sensors [132].

### 7.5. Surface Electromyography Based Control

Devices available commercially employ surface electromyography sensors (sEMG). These sensors detect EMG signals transmitted through the forearm, wrist, and hand. A neurological connection between the rehab device and the patient is created using these signals. sEMG sensors allow the devices to accurately measure and profile force and power transmitted by each hand. sEMG sensors provide great readings for muscular rehabilitation. Joan Lobo-Prat proved that a powered device could be directly controlled using sEMG control [118]. This study measured the success of EMG control on patients with Duchenne muscular dystrophy (DMD), and it would not apply to stroke victims [118]. EMG control has helped patients with spinal cord injuries or strokes by considering the muscular spasticity typical of stroke patients [133,134]. sEMG sensors can accurately predict movement and pain if the EMG signals are calibrated to read low-level signals [135]. An example of an EMG’s control algorithm is provided in Figure 14 from Wege’s EMG rehabilitation device.

The human hand uses several groups of muscles to extend and flex the appendages. Early devices used a binary switch to actuate the prosthetic, turning on when the EMG sensors received the signal from the arm [129]. Since EMG signals are being sent through several muscles, filtering the noise becomes essential to correctly drive the device by user intention. To circumvent the issue, researchers created multiple EMG inputs to track a pattern of motion [136]. This, in turn, helped create pattern recognition, as certain signals will indicate a grasping motion, gripping motion, etc. The biggest drawback of pattern recognition is the delay in response. While EMG signals are traveling through the hand, several variables need to be identified before matching to the correct motion. While the device processes the data, the user could be experiencing muscular spasticity or a new intention in motion. Instead, regression models for pattern recognition allow the control over the actuation to be conducted by two controllers, one for real-time movement and a second for updated prediction of the motion [129,137].

### 7.6. Force Sensor-Based Control

Force sensors are the most popularly employed control system. Since the patient will always be exerting pressure on the device to actuate the fingers, force sensors provide a natural reading of the patient’s motion. Using a combination of force sensors and other devices to detect position, such as Hall Sensors or potentiometers, the control system becomes highly accurate.

Devices such as the ExoHand, X-Glove, and HandMATE employ force sensors in conjunction with potentiometers and force-sensitive resistors to measure the input force from the patients [41,138]. Once the gloves have collected the input force from the patient, the feedback controllers employ an admittance control strategy. This combination creates a force feedback system, allowing the gloves to provide haptic feedback for the user [41]. Other devices such as the SEM glove and CyberGrasp utilize additional control strategies to provide a more precise feedback system. For the SEM glove, a PID controller is connected to the linear actuators of the device, creating smoother motion and somewhat adaptive control [41,54]. Since the CyberGrasp uses virtual reality to allow patients a haptic system, the controller must employ PID and admittance control for accurate angle and force measurement [28].

In addition, most devices use a closed-loop controller to predict force-sensing accurately [133,139]. Combining the physical feeling of movement with the assistive robotic device makes the patient more comfortable using the device [140]. Shown in Figure 15 is the force-sensitive control algorithm for the HandMATE. Figure 16 shows the force-feedback position from the measurement of one of the force sensors of the HandCARE device. Here, the motor current necessary to sustain a constant position of one finger, in this case the middle finger, when varying forces are applied is shown. The theoretical value would be a linear relationship, and the results of the experiment show that the relationship is aligned with the theoretical values.

## 8. Commercialization & Usability

Several devices have already been produced for commercial use, either in a clinic or at home. With the market share of assistive devices increasing, it is important to discuss the commercialization of these devices and their usability.

As technology grows, interfaces that allow for smoother user control are imperative. To provide standalone service with minimal assistance from an external source, patients seeking telerehabilitation or at-home solutions need a smart and intuitive interface. Amadeo and HandyRehab employ mobile apps and video games to encourage patients to exercise more. The skill-based exercises have been proven to improve motor function and increase patients’ focus on rehabilitation [83]. A simple interface is shown by the HandMATE, where an Android app is programmed to send and receive information regarding the current therapy mode selected by the patient and the movement data associated with each exercise [45]. Moreover, an advanced form could be taken by the CyberGrasp glove. The glove works with a virtual reality environment in this device, engaging patients while also providing haptic feedback [49].

Amadeo, Gloreha, and Saebo are a few companies that have created several commercial devices for a range of rehabilitation. Amadeo has created devices for lower extremity rehabilitation while also providing EMG-based control and smart interfaces [97]. Saebo, although not providing any powered devices, has a range of medical orthoses and supports for the upper and lower extremities; however, despite being very close in USD pricing, these devices do not incorporate any smart interfaces. Instead, Saebo provides a simple remote to track the position and movement of their devices [141].

## 9. Discussion

After reviewing and classifying the varying forms of HRDs, a newer idealized model of an HRD can be inferred. First, the correct class of HRD must be chosen. Orthoses, exoskeletons, and end-effector devices provide different balances of rehabilitation benefits. Newer designs should combine the advantages of each into one device. Future research will need to be conducted as there is no uniform scale to judge rehabilitation devices. The analysis of this paper has focused the discussion on clinical trials of HRDs rather than simulated results. Results from patients provide a better understanding of a device’s efficacy. Thus, the idealized model of an HRD should be based on clinical trials with FMA score improvements.

Orthoses primarily offer strictly active rehabilitation therapy, with some devices providing additional components to allow for resistive training. However, these braces’ low profile and costs inhibit them from competing with more developed exoskeleton devices. Exoskeletons provide the perfect combination of power and support. While the outer casing can be printed from low-cost plastics, exoskeletons can also be developed at high costs depending on the material needs. These devices provide all rehabilitation therapy exercises. Patients using exoskeletons report a greater improvement in their motor function recovery, as shown by the FMA, MAS, and CMSA scales. End-effector devices provide the same benefits while incorporating higher sensing capabilities and games to engage the patients. Although end-effector devices result in slightly higher patient scores, they are limited to a singular location and cannot be portable. Therefore, an exoskeleton device is preferred.

As shown by the mathematical models, the actuation of the HRD does not have strict requirements. Using different combinations of gears or links, any exoskeleton or end-effector device can properly actuate the flexion and extension of the hand. Furthermore, the only limitation on actuation is the weight and cost of the HRD. Preferably, linkage-based or cable-driven systems would be employed to provide the best torque to weight ratio while still attaining high precision and accuracy.

Several therapeutic strategies can provide key advantages for each rehabilitation device. Since the performance of the patients during the exercise is the most integral part, challenge-based and assistive strategies should be employed. With the highest FMA score improvements being recorded with devices that employ haptic and coaching strategies, future developments of HRDs should integrate these strategies. Furthermore, devices should respond naturally to patients, so sEMG control is crucial. Neuroplasticity is dependent on the patient equating the device’s motions to the neurological signals being sent and received. Incorporating force feedback would benefit the patient in this process. Therefore, the best control method to implement would be a combination of sEMG and force feedback controls.

Interfaces greatly improve the patient’s engagement with the device. Most humans already struggle with basic chores and exercise, so rehabilitation therapy is often difficult since patients find issues motivating the exercise. Incorporating virtual reality, exercise games, and self-driven ADL exercises engaged more patients in the rehabilitation process. Telerehabilitation allows patients to receive proper rehabilitation without being inhibited by travel, wait times, or availability.

An idealized HRD would therefore be an exoskeleton device actuated by either linear actuators or rotary motors. This device would also incorporate an assistive and challenge-based control strategy with force feedback. This encompassing model will allow the device to incorporate ADL exercises, games, and progress tracking in an easy-to-use interface.

## 10. Conclusions

After reviewing several commercial and documented rehabilitation devices, the need for improvement is still clear. Patients in need of HRDs have dynamic demands, since each patient would recover in unique ways. There is no rehabilitation device capable of operating passive, active, and assisted physical therapies for all forms and stages of hand rehabilitation. The use of sEMG sensors is not widely used, despite its direct connection to neuroplasticity and the recovery progress. Further, advanced therapeutic strategies have been proven to aid patients. These strategies require advanced technology, such as sEMG sensors, user feedback, and smart interfaces. The only standard in hand rehabilitation is the categorization of the injury’s severity using the CMSA scale. Without proper standards for rehabilitation progress, HRDs do not have set goals or uses. This, in turn, creates tangents in the HRD design and development. Future research must advance and focus development to a standardized form for hand rehabilitation.

## Figures and Tables

**Figure 1 micromachines-13-01033-f001:**
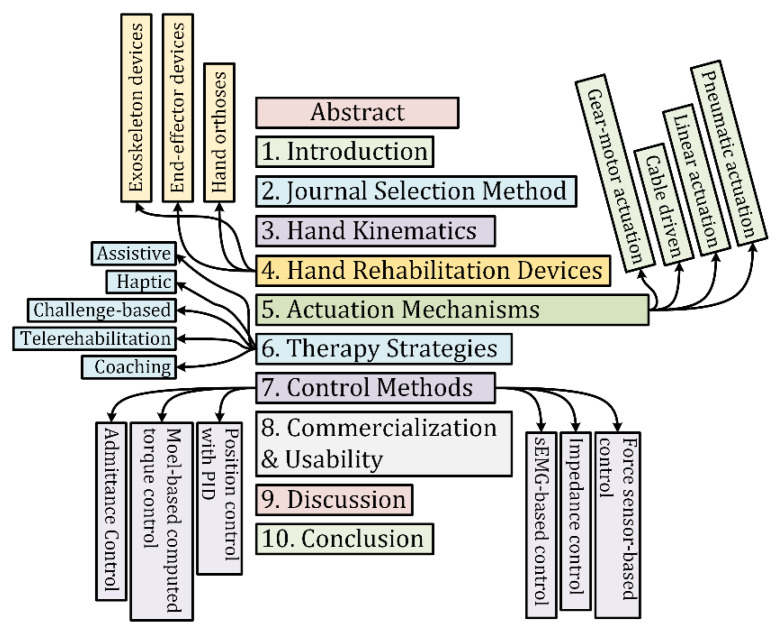
Organization of the Manuscript.

**Figure 2 micromachines-13-01033-f002:**
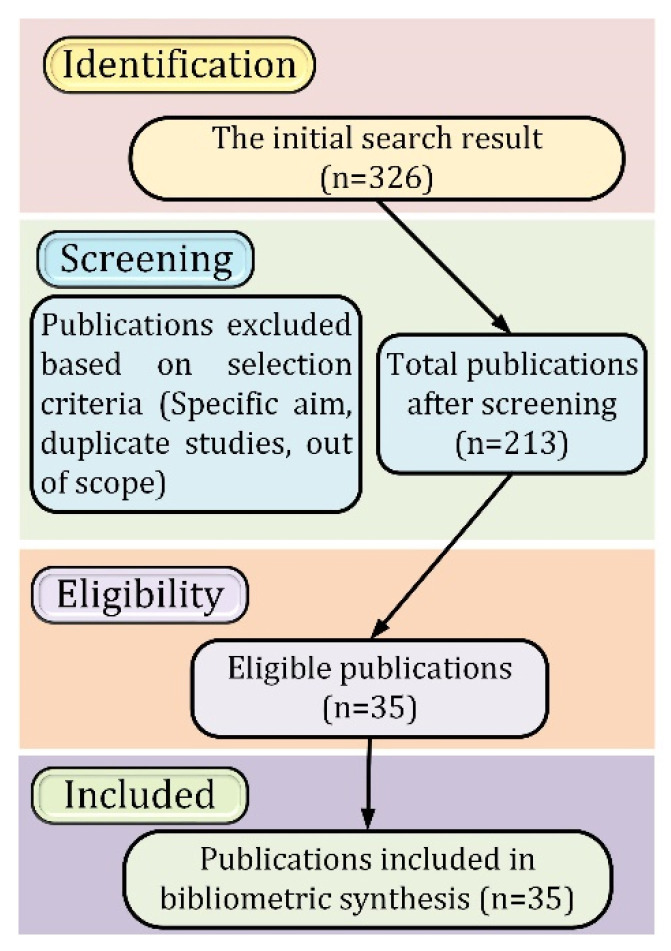
Inclusion and exclusion criteria of the selected studies.

**Figure 3 micromachines-13-01033-f003:**
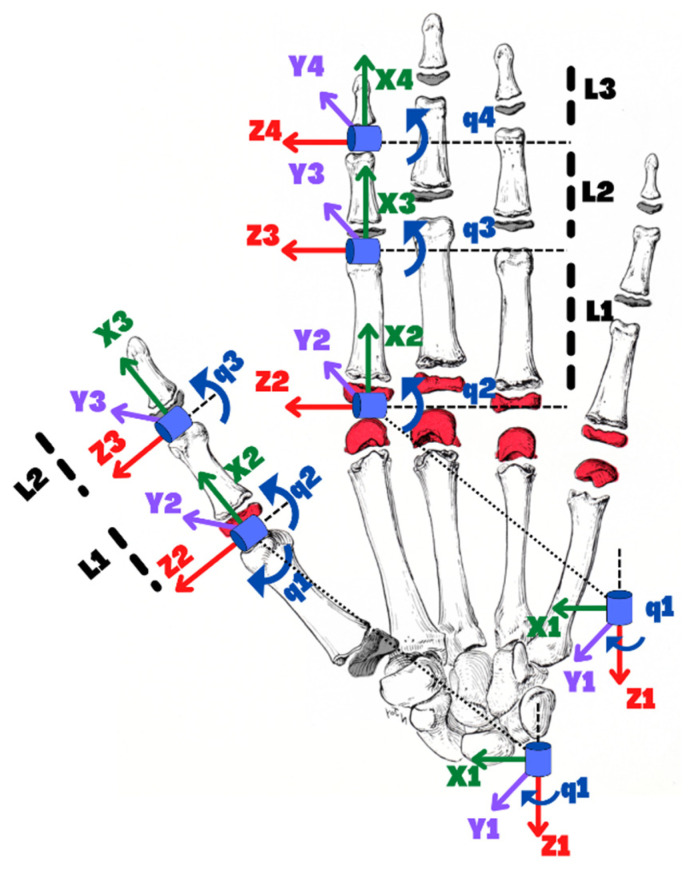
Reference Hand for Modified DH parameters.

**Figure 4 micromachines-13-01033-f004:**
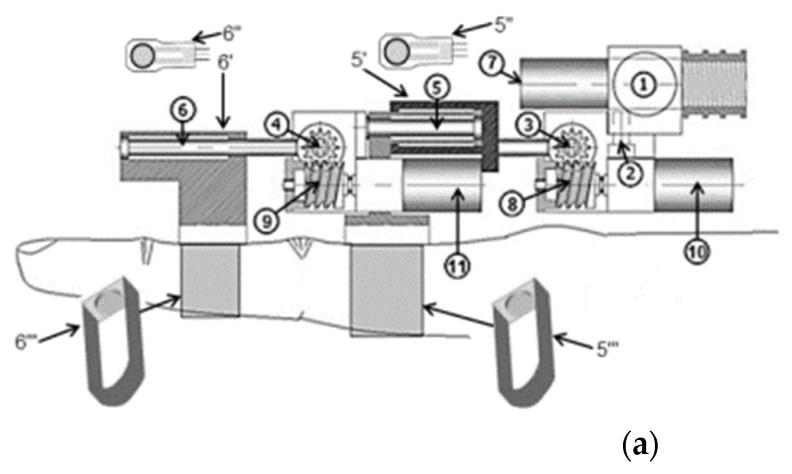
Mechanical designs for an exoskeleton and end-effector device. (**a**) the mechanical design of the ExoK’ab exoskeleton device (**b**) the mechanical design of the HapticKnob end-effector device [42,43].

**Figure 5 micromachines-13-01033-f005:**
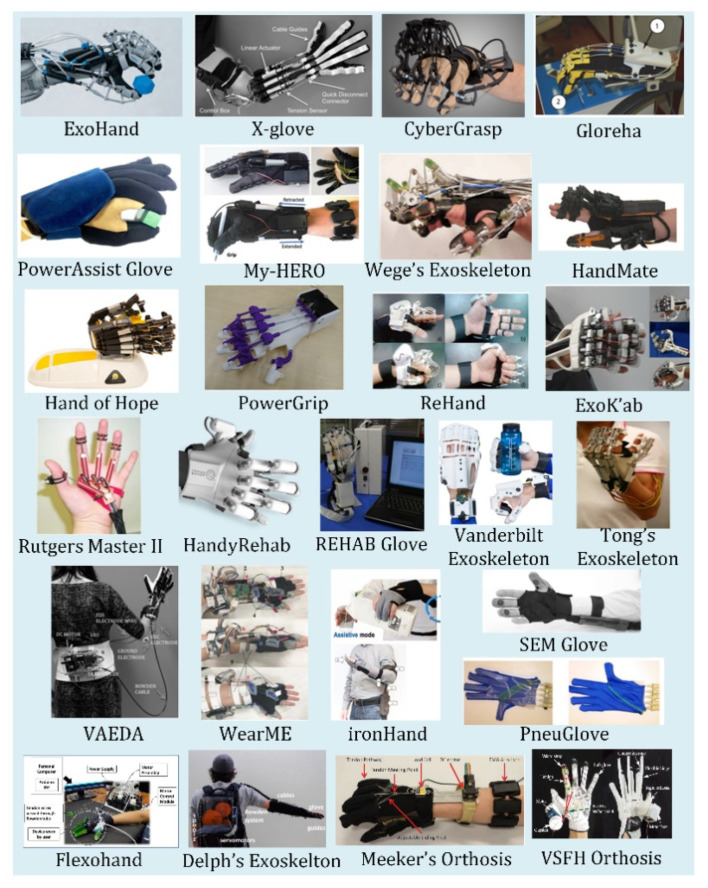
Exoskeleton Devices reviewed in this paper.

**Figure 6 micromachines-13-01033-f006:**
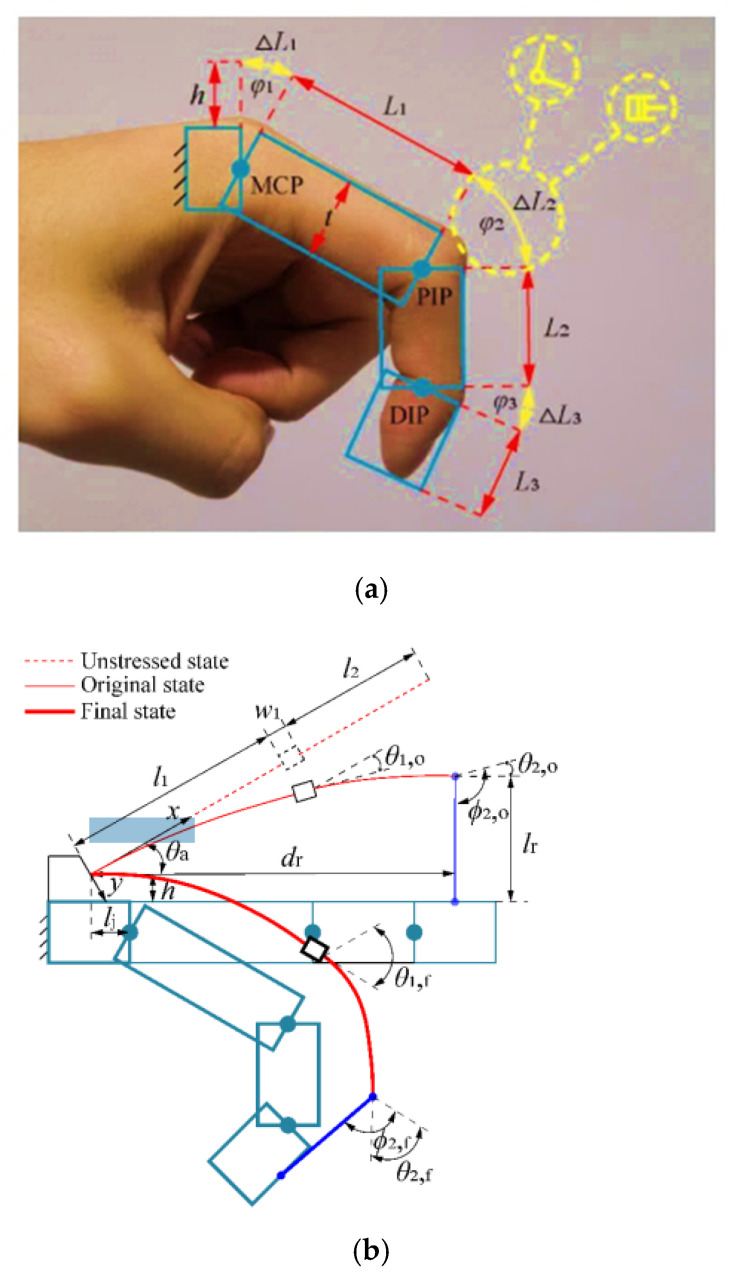
The base design principle for the hybrid-driven compliant hand exoskeleton [71]. (**a**) Basic design principle of the hand exoskeleton. (**b**) Human-machine coupling model based on human finger and the VSFH device. Here, $l_{i}$ represents the distance between the joints, $l_{r}$ is the length of the connecting rope from the VSFH device to the finger, $d_{r}$ is the horizontal distance between the connecting rope and fixed end of the VSFH device, $\theta_{a}$ is the angle between the VSFH and the horizontal line, and $w_{1}$ represents the length of the connecting block.

**Figure 7 micromachines-13-01033-f007:**
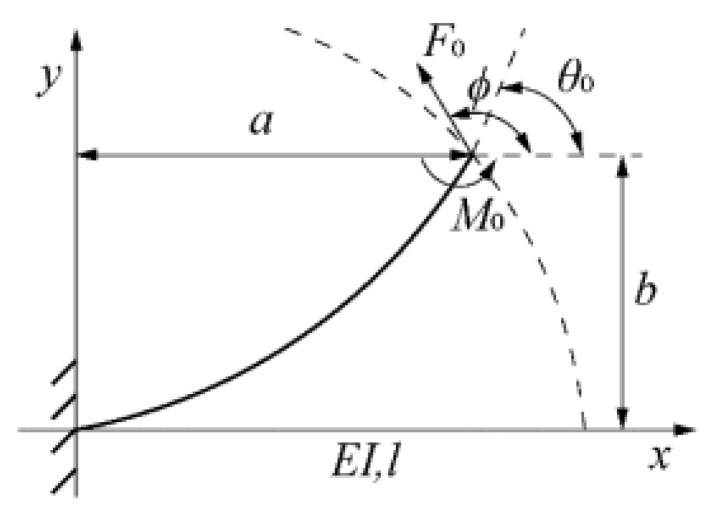
Shown is the deflection of a singular link on Meng’s design. Each joint can be modeled as a cantilever beam, and each initial position, torque, and angle can be adjusted accordingly.

**Figure 8 micromachines-13-01033-f008:**
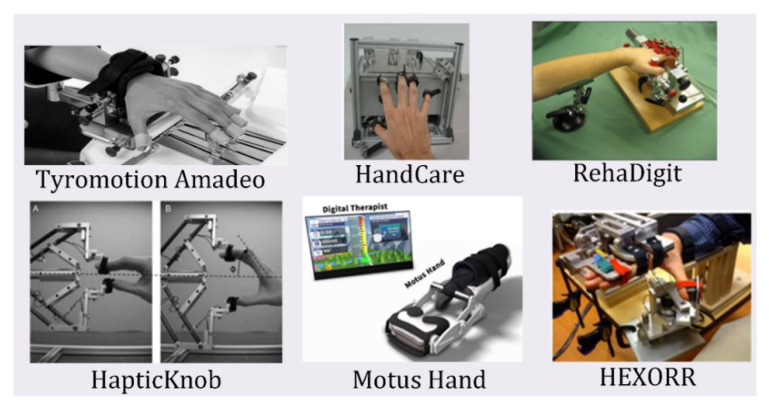
End-effector devices are reviewed in this paper. These devices use end-point control to actuate the hand for proper rehabilitation.

**Figure 10 micromachines-13-01033-f010:**
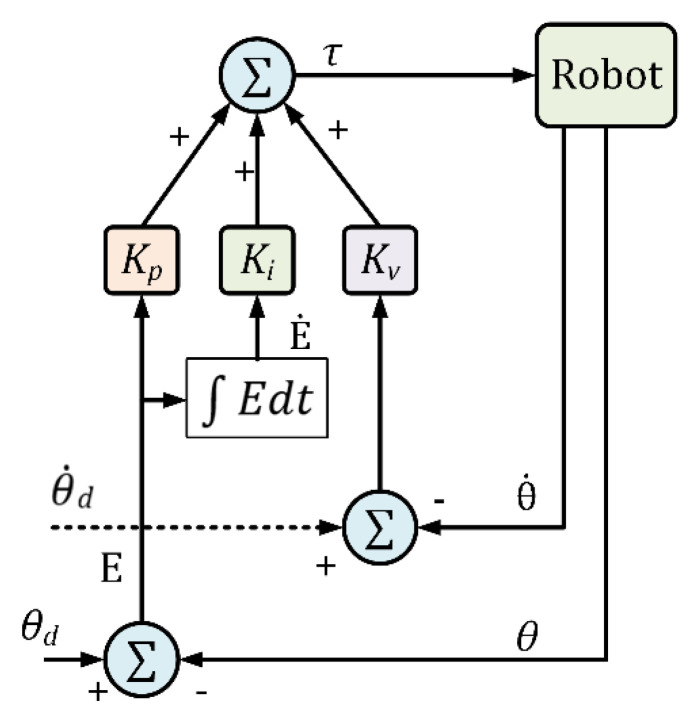
PID control schematic.

**Figure 11 micromachines-13-01033-f011:**
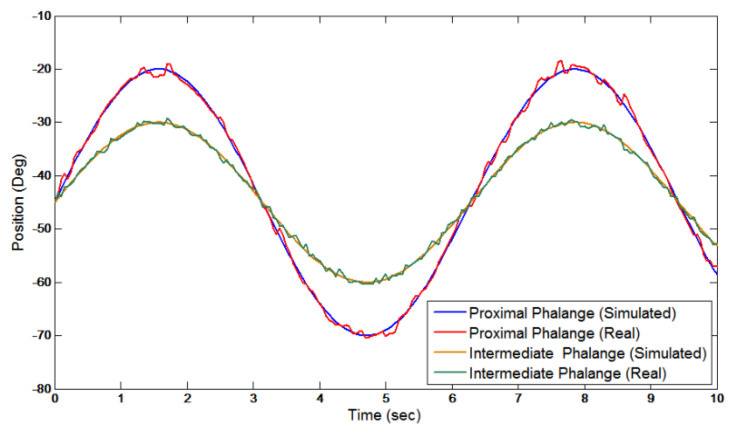
Simulated and actual trajectories of ExoK’ab device conducting a passive exercise.

**Figure 12 micromachines-13-01033-f012:**
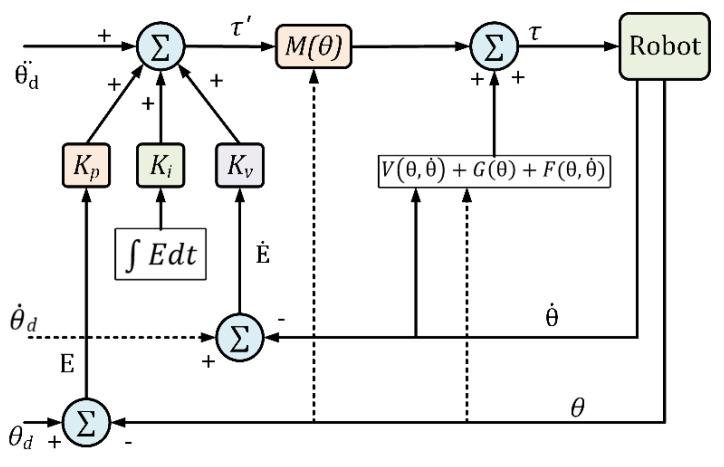
Computed torque control schematic.

**Figure 13 micromachines-13-01033-f013:**
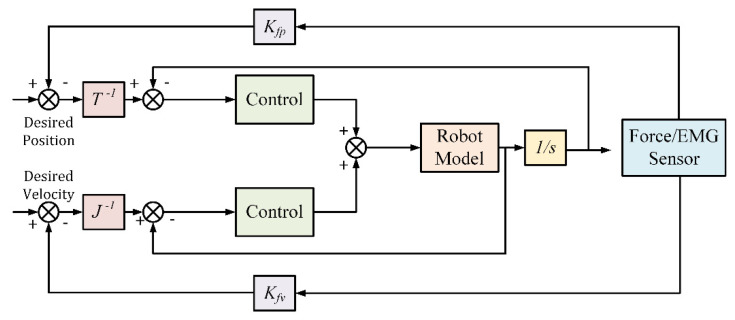
Sample Impedance Control schematic.

**Figure 14 micromachines-13-01033-f014:**
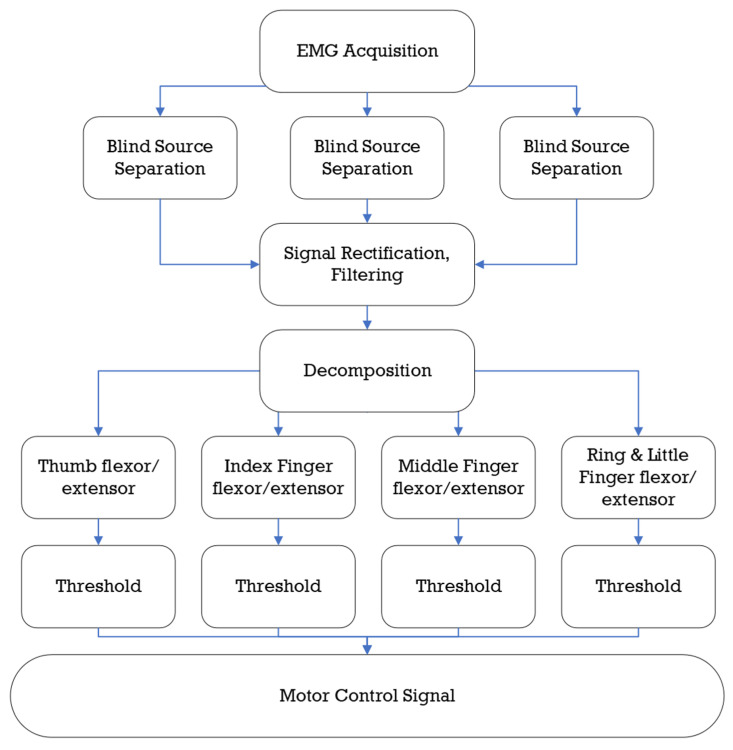
Wege’s EMG control algorithm recreated; The device begins by reading the sensed values from three sources before filtering and deciphering the data to control the actuation [70].

**Figure 15 micromachines-13-01033-f015:**
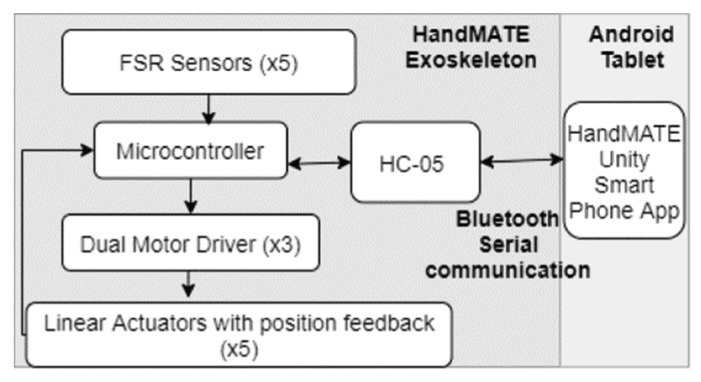
HandMATE’s control algorithm; The device uses the Force Sensitive Resistors (FSRs) at each joint to decipher the input before assisting in completing the exercise [45].

**Figure 16 micromachines-13-01033-f016:**
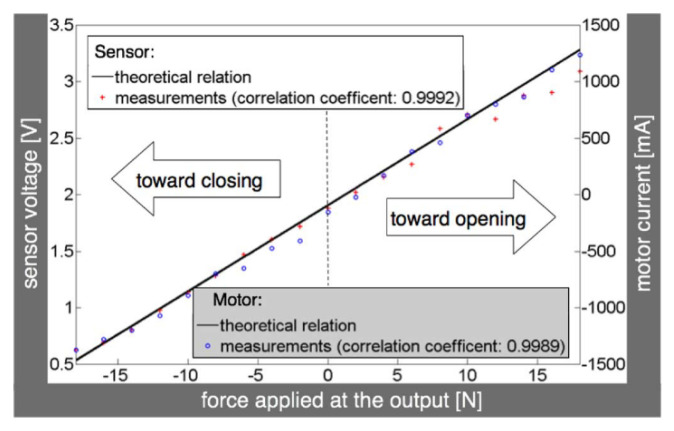
Theoretical and actual measurements of motor current required to maintain the position of a chosen finger, in this case the middle finger [64].

**Table 1 micromachines-13-01033-t001:** Modified DH parameters for all fingers, excluding the thumb.

Link	$\alpha_{i-1}$	$a_{i-1}$	$d_{i-1}$	$\theta_{i}$	Joint Axis Motions
**1**	0	0	0	$q_{1}$	MCP abd./add.
**2**	$\frac{\pi }{2}$	0	0	$q_{2}$	MCP flex./ext.
**3**	0	$L_{1}$	0	$q_{3}$	PIP flex./ext.
**4**	0	$L_{2}$	0	$q_{4}$	DIP flex./ext.
**Fingertip**	0	$L_{3}$	0	0	-

**Table 2 micromachines-13-01033-t002:** Modified DH parameters for Thumb.

Link	$\alpha_{i-1}$	$a_{i-1}$	$d_{i-1}$	$\theta_{i}$	Joint Axis Motions
**1**	0	0	0	$q_{1}$	MCP abd./add.
**2**	$\frac{\pi }{2}$	0	0	$q_{2}$	MCP flex./ext.
**3**	0	$L_{1}$	0	$q_{3}$	DIP flex./ext.
**Fingertip**	0	$L_{2}$	0	0	-

**Table 3 micromachines-13-01033-t003:** Healthy ROM for each joint and finger.

		Flexion (°)	Extension (°)	Abduction/Adduction (°)
Thumb	MCP	75–80	0	--
	DIP	75–80	5–10	--
Index	MCP	90	30–40	60
	PIP	110	0	--
	DIP	80–90	5	--
Middle	MCP	90	30–40	45
	PIP	110	0	--
	DIP	80–90	5	--
Ring	MCP	90	30–40	45
	PIP	120	0	--
	DIP	80–90	5	--
Little	MCP	90	30–40	50
	PIP	135	0	--
	DIP	90	5	--

**Table 4 micromachines-13-01033-t004:** Measured exerted forces on pathological hands.

Patient	1		2		3		4	
Measured Value (N)	Flexion	Extension	Flexion	Extension	Flexion	Extension	Flexion	Extension
**Thumb**	7.0	6.5	3.0	1.9	5.2	5.5	7.0	5.5
**Index Finger**	9.5	5.0	2.0	2.0	2.8	2.6	4.5	1.6
**Middle Finger**	7.5	5.3	1.6	1.9	4.0	6.0	6.0	2.8
**Ring Finger**	8.0	9.0	2.6	1.5	4.5	7.0	6.0	2.8
**Little Finger**	7.0	3.4	3.0	1.4	5.2	9.0	7.0	5.0

**Table 5 micromachines-13-01033-t005:** Actuated Devices with sensors and a brief description.

Name	Description
**HandMATE** [45]	HandMATE is an updated design from the HandSOME II where the authors have replaced the non-motorized components with a linear actuated, linkage-based system. Using force-sensing resistors (FSRs), this exoskeleton device can actuate over the MCP, PIP, and DIP joints.
**Tyromotion Amadeo** [46]	Tyromotion’s Amadeo device is an end-effector device capable of actuating each individual finger. Using EMG sensors and a linear rail system, the Amadeo has proven to improve the FMA scores of several patients
**Gloreha** [36]	Gloreha is a pneumatically actuated exoskeleton that actuates each finger using a cable-driven system. This device has proven to improve the FMA scores of patients.
**X-GLOVE** [47]	The X-Glove is an exoskeleton capable of actuation over the hand’s joints using linear actuation and a linkage-based system. The glove is capable of modest FMA score improvement by employing tension sensors in the device
**ExoHand** [48]	Festo’s Exohand is a linkage-based exoskeleton that employs linear actuators to achieve rotation of the MCP, PIP, and DIP joints. Using force and tension sensors, the device provides controlled passive and active rehabilitation
**CyberGrasp** [49]	CyberGrasp is an intricate exoskeleton capable of VR rehabilitation and actuation. This glove implements linear actuation via a linkage-based system and FSRs to provide safe rehabilitation
**HEXORR** [50]	HEXORR is an end-effector device that provides hand rehabilitation using brushless DC motors and a linkage-based system. To provide safe control, torque sensors are employed
**Rutgers Master II** [51]	A modified Rutgers Masters II device has been reviewed to determine the efficacy of the pneumatically actuated exoskeleton device. Implementing tele-rehabilitation and EMG sensors, the modified device can acutely improve FMA scores
**PowerAssist Glove** [52]	The Power Assist Glove is a pneumatically actuated exoskeleton device. Using pressure sensors in each finger, the device is capable of rehabilitation for the hand
**REHAB Glove/Hagshenas-Jaryani** [53]	The REHAB glove is an exoskeleton device that implements pneumatic actuation. A combination of inertial and pressure sensors allows the device to provide rehabilitation
**SEM Glove** [54]	The SEM Glove is a cable-driven exoskeleton device. With FSRs and linear actuators, this device provides passive rehabilitation
**IronHand** [55]	The IronHand exoskeleton device is a pneumatically actuated device. Using FSRs, this device can provide rehabilitation with modest improvements in hand strength
**Hand of Hope** [56]	Hand of Hope is a linearly actuated device that utilizes a linkage-based system. The EMG sensors provide helpful data for the passive, active, and assisted rehabilitation exercises that are provided by the device.
**WearMe** [57]	WearMe is a cable-actuated exoskeleton device that is capable of rehabilitation. This device implements inertial movement sensors to provide safe rehabilitation.
**Exo Glove** [58]	The Exo Glove is a linearly actuated exoskeleton device that uses a linkage-based system to provide safe actuation. Using FSRs and EMG, the device provides safe rehabilitation
**Motus Hand** [59]	The MOTUS hand is a pneumatically actuated exoskeleton device. Using a user interface and kinematic and EMG sensors, this device can provide safe rehabilitation and is the only FDA Class 1 at-home rehabilitation device that provides active-assistance
**My-HERO** [60]	My-HERO is an improvement on the HERO design with the addition of EMG sensors. This exoskeleton device implements a cable-driven actuation system using linear actuators.
**RehaDigit** [61]	The Reha-Digit is a unique end-effector device that uses cylinders to roll the fingers for actuation over the MCP, PIP, and DIP joints. Using DC motors, this device boasts modest FMA score improvements
**PowerGrip** [62]	The PowerGrip exoskeleton device provides safe actuation over the hand using a linkage-based system and high-torque servo motors
**Vanderbilt** [63]	The Vanderbilt exoskeleton device utilizes brushless DC motors and a linkage-based system to provide safe actuation over the hand. This device uses Hall sensors in conjunction with the motors for safe actuation
**HandCARE** [64]	The HandCARE end-effector device is a cable-driven device that provides rehabilitation using brushless DC motors. Using FSRs, the device can control the rehabilitation precisely
**HandyRehab** [65]	HandyRehab is a new exoskeleton device that uses linear actuators and a linkage-based system to provide safe rehabilitation. This device implements a user interface and EMG sensors to provide accurate rehabilitation for the patient.
**VAEDA** [66]	VAEDA is a voice-activated exoskeleton device capable of modest FMA score improvement. Using brushed DC motors and a cable-driven system in conjunction with the EMG and tension sensors, the device provides safe rehabilitation for patients
**ReHand** [67]	The ReHand is an exoskeleton device that uses gear motors and a linkage-based system to provide a small FMA score improvement. Using EMG sensors and voice control, this device is capable of safe rehabilitation
**PneuGlove** [68]	The PneuGlove is a pneumatically actuated device capable of modest FMA score improvement. Implementing force bend sensors allows the device to provide safe control
**Tong et al.** [69]	Tong et al.’s exoskeleton device uses a linkage-based system and linear actuators to provide safe passive, active, and assistive rehabilitation. Using EMG sensors, this device has precise control over the hand
**Haptic Knob** [43]	The Haptic Knob is an end-effector device capable of moderate FMA score improvement. Using a pulley and linkage-based system with the brushed DC motors, this device implements force sensors for precise movement
**Wege’s device** [70]	Wege et al.’s exoskeleton device utilizes brushless DC motors and a cable-driven system to provide safe passive rehabilitation. A combination of force and EMG sensors allows the device to improve a patient’s strength
**ExoK’ab** [42]	The ExoK’ab is an exoskeleton device capable of accurate passive rehabilitation. Using force sensors in conjunction with the gearmotors and gear trains, this device actuated over each joint accurately.
**VSFH** [71]	VSFH is an exoskeleton device designed with a variable stiffness in each joint. Using FSRs and a cable-driven system, the VSFH device can provide passive rehabilitation and conduct common ADLs
**Meeker et al.** [72]	Meeker et. al. developed an exoskeleton device capable of passive rehabilitation. Using gearmotors and a cable-driven system, Meeker’s device employs EMG sensors to provide safe actuation
**Flexohand** [73]	Flexohand is the only exoskeleton device capable of independent actuation of each MCP, PIP, and DIP rotation. Using gear motor servos with a cable-driven system, passive rehabilitation is conducted
**Delph et al.** [74]	Delph et al. developed an exoskeleton device capable of passive, active, and resistive rehabilitation using servo motors and a cable-driven system. Using EMG and tension sensors, this device can assist or resist a user’s motion and provide safe actuation

**Table 7 micromachines-13-01033-t007:** Powered Devices classified into actuation mechanisms.

Name	Exoskeleton	End-Effector	Actuator	Actuation Mechanism
**HandMATE**	√	-	Linear Actuator	Linkage system
**Tyromotion Amadeo**	-	√	Linear Actuator	Linear Rails
**Gloreha**	√	-	Pneumatic Piston	Cable-driven
**X-GLOVE**	√	-	Linear Actuator	Linkage system
**ExoHand**			Linear Actuator	Linkage system
**CyberGrasp**	√	-	Linear Actuator	Linkage system
**HEXORR**	-	√	Brushless DC Motor	Linkage system
**Rutgers Master II**	√			
**PowerAssist Glove**	√		Pneumatic Actuator	Pneumatic Tubes
**REHAB Glove**	√		Pneumatic Actuator	Pneumatic Tubes
**SEM Glove**	√	-	Linear Actuator	Cable-driven
**IronHand**				
**Hand of Hope**	√	-	Linear Actuator	Linkage system
**WearME**	√	-	Brushless DC Motor	Cable-driven
**Exo Glove**	√	-	Linear Actuator	Linkage system
**Motus Hand**	√	-	Pneumatic Actuator	Linkage system
**My-HERO**	√	-	Linear Actuator	Cable-driven
**RehaDigit**	-	√	DC Motor	-
**PowerGrip**	√	-	Servo Motor	Linkage system
**Vanderbilt**			Brushless Motor	Linkage system
**HandCARE**	-	√	Brushless DC Motor	Cable-driven
**HandyRehab**	√	-	Linear Actuator	Linkage system
**VAEDA**	√	-	Brushed DC Motor	Cable-driven
**ReHand**	√	-	DC gear motor	Linkage system
**PneuGlove**	√	-	Pneumatic Actuator	Pneumatic Tubes
**Tong et. al**	√	-	Linear Actuator	Linkage system
**Haptic Knob**	-	√	Brushed DC Motor	Pulley and linkage system
**Wege’s Exoskeleton**	√	-	Brushless DC Motor	Cable-driven
**ExoK’ab**	√	-	Brushless DC gear motor	Gearmotor
**Meeker**	√	-	Brushed DC gear motor	Cable-driven
**VSFH**	√		Servo Motor	Cable-driven
**Delph**	√	-	Servo Motor	Cable-driven

**Table 8 micromachines-13-01033-t008:** Comparison of Rehabilitation Strategies and FMA score improvements.

Device	FMA Improvement	Assistive	Challenge Based	Haptic	Coaching	Telerehabiltiation
**Tyromotion Amadeo** [80]	**7**	√	√		√	
**Gloreha** [79]	**8**	√	√		√	√
**Hand of Hope** [83]	**4**	√	√	√	√	√
**My-HERO** [60]	**8.4**	√			√	
**ReHand** [67]	**2**	√			√	
**VAEDA** [66]	**2**	√	√			
**Rutgers Master II** [96]	**2**		√	√		√
**X-GLOVE** [47]	**4**	√			√	
**HEXORR** [78]	**2**	√	√			
**Haptic Knob** [82]	**3**	√		√		
**RehaDigit** [61]	**1.2–6**	√				
**PneuGlove** [68]	**4**			√		

## Data Availability

No new data were created or analyzed in this study. Data sharing is not applicable to this article.

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
