# Peer review of "Hand Rehabilitation Devices: A Comprehensive Systematic Review"

_micromachines, 2022, doi:10.3390/mi13071033_

Round 1

Reviewer 1 Report

The current paper proposes a review for hand rehabilitation devices processes. study evaluates rehabilitation devices designed for the hand's passive, active, and active-assisted physical therapy. The devices are compared by their mechanical designs, actuation mechanisms, control systems, and interfaces to determine a newer idealized model.

Comments to authors:

- The theoretical part of this paper is inexistent. Maybe it would be better than a mathematical model should be added.

- The authors use over 100 paperers to write this paper, but noting concluding is said. Probably a comparison between the mathematical models of exoskeleton devices would be relevant.

- Probably would be better to add a comparison study the control algorithms that have been applied to this kind of process.

- Maybe it would be a plus if the authors would compare the experimental or simulation results obtained for this type of process.

Author Response

Please see the attached reviewer response

Thank You

Reviewer 2 Report

1. The article described hand rehabilitation equipment, but only 33 HRD were screened out (34 in the abstract, 33 in Figure 2). This paper is just a classification list of the literature and introduces the devices of each type of HDR, but does not make clear the history, technical pain points and future development direction of HRD. It is suggested to describe the development of HRD technology and the development of future technology.

2. HRD is divided into Orthoses, Exoskeletons and end-effector devices based on the availability of power sources. In addition, according to the article, orthoses can provide active or resistive training, while in Table 1, orthoses provide active, resistive and passive training. Motor assist is also mentioned in Table 1. Does it mean the orthoses have power? What's the difference between Orthoses and Exoskeletons and end-effector devices? Generally speaking, the classification of HRD is not clear.

3. The abstract of the article mentions the new Model twice, but there is no description of this new model in the body. What is the significance of this new Model?

4. From the abstract, the third part mainly talks about the mechanical designs of HDR equipment, but the article only describes the equipment and functions, and does not involve the mechanical designs. And in 3.2 Exoskeletons and End-Effector Devices, only the End-Effector device is covered.

5. In Figure 9, HRD review is written in the article, but wrist and Elbow Rehabilitation device appear, which have no effect on the article.

Author Response

(The authors gave the same response as above.)

Reviewer 3 Report

I would recommend, if possible, adding other topical references, for example:

In line 376: “Rehabilitation has now incorporated videogames and exercises to engage patients further in the rehabilitation progress” [Alarcón-Aldana, A.; Callejas-Cuervo, M.; Bo, A. Upper Limb Physical Rehabilitation Using Serious Videogames and Motion Capture Systems: A Systematic Review. Sensors 202020, 5989]

In line: 316: Both end-effector and exoskeleton devices can employ several control strategies using varying combinations of sensors” [VélezGuerrero, M.A.; Callejas-Cuervo, M.; Mazzoleni, S. Design, Development, and Testing of an Intelligent Wearable Robotic Exoskeleton Prototype for Upper Limb Rehabilitation. Sensors 202121, 5411].

Author Response

(The authors gave the same response as above.)

Round 2

Reviewer 1 Report

The authors improved the paper since the last version, but the theoretical part is still inexistent and the authors should improve the paper considering my previous comments.

Author Response

(The authors gave the same response as above.)

Reviewer 2 Report

  • The author addressed all the questions

Author Response

Thank You.

Round 3

Reviewer 1 Report

The paper still have major lacks, see the previous comments.